# A Study of Frailty, Mortality, and Health Depreciation Factors in Older Adults

**DOI:** 10.3390/ijerph17010211

**Published:** 2019-12-27

**Authors:** Jwu-Rong Lin, Erin Hui-Chuan Kao, Shuo-Chun Weng, Ellen Rouyer

**Affiliations:** 1Department of International Business, Tung Hai University, No.1727, Sec.4, Taiwan Boulevard, Xitun District, Taichung 40704, Taiwan; jrlin@thu.edu.tw (J.-R.L.); erinkao@thu.edu.tw (E.H.-C.K.); 2Institute of Clinical Medicine, National Yang-Ming University, No. 155, Section 2, Linong St, Beitou District, Taipei City 112, Taiwan; j01i02m3@yahoo.com.tw; 3Center for Geriatrics and Gerontology, Division of Nephrology, Department of Internal Medicine, No. 1650, Section 4, Taiwan Boulevard, Xitun District, Taichung 40705, Taiwan

**Keywords:** frailty, mortality, health depreciation factors, recursive probit regression model

## Abstract

This study used 23 factors (eight interval variables and 15 dummy variables) as proxies for health depreciation. We used 1248 older adults from the Center for Geriatrics and Gerontology of Taichung Veterans General Hospital (Taiwan) to examine the association among frailty, health depreciation, and mortality in older adults. This study found that a significant positive correlation existed between frailty and mortality in older adults. Further, we applied a recursive bivariate probit model to examine the association between health depreciation factors, frailty, and mortality. Our results showed that health depreciation factors, such as Charlson’s comorbidity index, diabetes and hyperlipidemia, significantly increased older adults’ frailty; in contrast, albumin and mini nutritional assessment significantly decreased older adults’ frailty. Through the frailty regression, we confirmed not only that health depreciation factors significantly influenced mortality, but also that creatinine, myocardial infarction, and malignant tumors could directly and significantly increase older adults’ mortality.

## 1. Introduction

At present, Taiwan is facing the problems of an aging population and a low birth rate. According to statistics released by the Ministry of the Interior (https://www.moi.gov.tw/stat/news_detail.aspx?sn=13742), the proportion of Taiwan’s population that is older than 65 increased from 6.81% in 1992 to 15% in 2012, exceeding the threshold of 7% used to define an aging society by the World Health Organization. The 2012–2060 Population Estimation in Taiwan report published by the Council for Economic Planning and Development, Executive Yuan (https://www.ndc.gov.tw/en/cp.aspx?n=2E5DCB04C64512CC), shows that the population of older adults in 2060 will be 2.9 times that in 2012; furthermore, the older population as a proportion of the total population will increase from 11.2% in 2012 to 39.4% in 2060. In addition, the old-age-dependency ratio will rise from 15% to 77.7%, and the aging index will sharply grow from 76.3 to 401.5. The aforementioned statistics reveal that Taiwan has become an aging society, and thus, establishing an elderly-care network is imperative. In addition, the Executive Yuan proposed the “10-Year Long-Term Care Project 2.0” in 2016 for an aging society. In order to implement the Long-Term Care Project successfully, understanding the elderly-care industry is a prerequisite.

Numerous countries have made the treatment of frailty (FR) the core issue in disability prevention and health care for older adults. Frailty is an expression of the common risk factors for diseases in older adults as well as chronic diseases (collectively referred to as HD—health depreciation—in this paper). To reduce resource depletion in long-term care services, this study examines the association between frailty (FR), mortality (MR), and health depreciation (HD) in older adults for the elderly-care industry and the government.

A literature review revealed that numerous measurements have been adopted to construct frailty (FR) indices. For example, Fried et al. [1] constructed an FR index that contains five dimensions for evaluating physical performance. Although their index fails to consider cognitive, emotional, psychological, social, and environmental factors, it can be easily used for objective evaluations, and therefore is still widely applied. This study applies the index established by Fried et al. [1] and has modified it according to the physical characteristics of Asian people (i.e., see Section 3.1). The participants were categorized as either FR, intermediate FR, and not FR. Compared to FR, we classify the last two categories (intermediate FR and not FR) as relatively healthy samples.

Fried et al. [1] proposed a hypothesis regarding the key factors resulting in FR, as shown in Figure 1. This study uses a recursive probit binary choice regression to investigate the impact of HD (e.g., chronic nutritional deficiencies, multiple chronic diseases, and pathological aging) on FR and MR in older adults.

In addition, this study employs a database containing the data of 1248 older adults provided by the Center for Geriatrics and Gerontology at Taichung Veterans General Hospital. Using the data, we establish a simultaneous recursive probit regression model to investigate the association between FR, MR, and HD in older adults. The method was applied as follows: (1) we construct both a theoretical foundation and an empirical model and consider variables based upon the perspective of medical economics; (2) according to the five symptoms of FR defined by Fried et al. [1], we categorize the participants’ symptoms, namely unintended weight loss, exhaustion, low activity, slowness, and low grip strength; those who had three or more symptoms were categorized in the FR group (N = 373), and those with 0–2 symptoms were categorized into the relatively healthy group (N = 875); (3) the correlation between FR and mortality and the difference test between three health-care types in terms of FR, MR, and HD are analyzed; and (4) we construct simultaneous recursive probit regression models to estimate the impact of various HD factors (e.g., hypertension, hyperlipidemia, hyperglycemia, tumors, pulmonary diseases, cerebrovascular diseases, blood creatinine, hemoglobin, and other chronic diseases) on the FR and MR in older adults.

Most prior empirical research employed the medical economics theoretical model to investigate the effects of key variables on older adults’ HD [2], FR, and health-care needs; these key variables include health-care quality, health-care price, wage rate, medical insurance, non-salary income, educational level, and age [2,3,4,5,6,7,8,9,10,11,12,13,14,15]. Most of these studies used the dataset constructed by public health systems, which are cross-sectional or panel data, but rarely employed microscopic data for each older adult, which have item-by-item health tests [1,4,16,17,18,19,20,21,22,23,24,25,26,27,28,29,30]. This study, directly using the data of older adults to examine the relationship between HD, FR, and MR, can provide more straightforward evidence for long term care projects, and bridges the gap in the literature. In addition, simultaneous recursive probit binary regression models are applied in this study, which differs from the conventional method of least squares and allows for a more robust empirical analysis.

The remainder of this paper is organized as follows: Section 2 introduces the theoretical framework and empirical model; Section 3 analyzes the empirical results; and Section 4 delineates the research conclusions and suggestions.

## 2. Theoretical Foundation and Empirical Model

To empirically investigate FR and MR in older adults, this study first establishes the theoretical foundation for our empirical model. The expenditure function for older adults is expressed as Equation (1):(1)$$\mathrm{EXP}\;=\;Q_{S}\times \frac{Q_{R}}{Q_{S}}\times P_{R}$$
where EXP denotes the total health-care expenditure during a hospital stay; Q_S_ denotes the amount of care service provided, measured in time, which is regarded as the substitution variable for older adults’ health-care needs; Q_R_ represents the amount of health-care resources invested; and Q_R_/Q_S_ represents the use efficiency of health-care resources (the reciprocal of productivity). A small Q_R_/Q_S_ indicates high health-care productivity, which results in superior health-care performance. The health-care price (P_R_) is influenced by factors such as the physical and mental states of older adults during their care period (referred to as HD in this paper) and personal or family conditions (e.g., height, weight, age, and sex). During older adults’ care period, interactions between Q_S_, Q_R_/Q_S_, and P_R_ increase the total expenditure, thereby facilitating the improvement of older adults’ health; in turn, this reduces the level of FR and increases the survival rate (SR). This can be shown in Equation (2):
FR = f(EXP) *f*′ < 0 *f*″ > 0SR = g(EXP) *g*′ > 0 *g*″ < 0(2)

According to Equation (2), a large EXP can slow the progression of FR in older adults and increase their SR (*f*′ < 0, *g*′ > 0); a marginal increase (*f*″ > 0) and decrease (*g*″ < 0) are expressed separately. This study adopts the theoretical foundation established using Equations (1) and (2) to construct an empirical model related to FR and MR for older adults (survival).

This study investigates the effect of HD on FR for older adults, who are taken care of in outpatient clinics, in hospitals or in care facilities. First, we employ a health production function to establish the key variables in the empirical model. A health production function refers to the maximum level of health an individual can achieve from specific medical care inputs within a given time period. From a mathematical perspective, this function represents the level of output (degree of FR) determined by the amounts of input (degree of HD). The older adult FR function could be established as follows:
FR = f (HD, CV)(3)
where FR is the indicator of frailty at a specific time point; HD includes eight interval variables (e.g., albumin) and 15 nominal data items (e.g., whether an older adult has diabetes [DM]); and includes sex as a control variable (CV) [31,32,33,34].

Following the same theoretical foundation as the health production function, this study subsequently explores the marginal influence of FR and HD factors on the post-discharge mortality rate (MR) in older adults during the study period. The setting for the empirical model was as follows [26,35,36]: MR = g (FR, HD, CV)(4)

In Equation (4), the dependent variable MR represents mortality (survival = 0, death = 1). Because only a 3-year follow-up was conducted and the number of individuals who died (N = 77) accounted for only 6.17% of the total population (N = 1248), this study uses the binary logistic model instead of the Cox proportional hazard model to estimate MR, where explanatory variables including FR, HD, and CV are considered.

On the basis of the models in Equations (3) and (4), this study constructs a logical framework for the simultaneous recursive probit model, as shown in Figure 2.

## 3. Empirical Results Analysis

### 3.1. Construction of the FR Index

This study constructed its FR index by modifying that developed by Fried et al. [1] according to the physical characteristics of the study population (i.e., elderly people in Taiwan), as shown in Table 1.

### 3.2. Sample Distribution in Terms of FR and DR

This study used 1248 valid samples from the Center for Geriatrics and Gerontology at Taichung Veterans General Hospital. The individuals were categorized into the FR group (FR ≥ 3, N = 373) and the relatively healthy group (FR = 0–2, N = 875) according to the five dimensions of the FR index in Table 1. During the 3-year follow-up period, the number of older adults who died was 77, and that of older adults who survived was 1171.

First, the correlation between FR and MR is analyzed. Table 2 reveals that the number and percentage of surviving older adults in the robust group are significantly higher than those of surviving older adults in the FR group (Pearson χ^2^(1) = 5.334, *p*-value = 0.021), which means the association between frailty (FR) and mortality (MR) is positive at a significance level of 5%.

### 3.3. Difference Analysis of Older Adults’ FR and HD

This study collected eight interval data items for HD (e.g., albumin (Alb)), 15 dummy variables (e.g., existence of DM), and one control variable (sex). Table 3 presents the mean, standard deviation (SD), valid samples, and difference test results for the FR and relatively healthy groups. The results of an unpaired *t* test (interval variables) and Mann–Whitney test (dummy variables) reveal that the HD factors, including albumin (Alb), fasting glucose (F_Glu), hemoglobin (HgB), mini nutritional assessment (TOT_5), age-adjusted Charlson comorbidity index (TOT_11), Diabetes (DM), hypertension (HT), hyperlipidemia (Hyper), congestive heart failure (CHF), cerebrovascular disease (CVAD), dementia, connective tissue disease (CTD), peptic ulcer disease (PUD), chronic kidney disease (CKD), hemiparesis (Hemi), and moderate-to-severe liver disease (L_disease) all have a significant influence on older adults’ FR. Furthermore, regarding the control variable, male older adults were healthier than female older adults.

### 3.4. Difference Analysis of Mortality and Health Depreciation in Older Adults

Panels A and B (HD factors) and Panel C (control variable) of Table 4 present statistical data concerning the older adults’ survival and mortality. The results of a *t* test on the interval variables of HD indicate that Alb, Cr, HgB, TOT_5, and TOT_11 significantly influence older adults’ MR. Furthermore, among the HD dummy variables, HT, Hyper, CHF, CVAD, PUD, CKD, Hemi, M_tumor, and L_disease all significantly influence the survival of older adults. Finally, MR does not significantly differ between older adults of different sex.

### 3.5. Parameter Estimation for the Recursive Probit Regression Model

According to Equations (3) and (4), this study employed a probit regression model and recursive probit regression model respectively to estimate the effects of HD factors on older adult’s FR and MR. Table 5 presents the results, which indicate the following:
The regression equations for individual estimations of FR and MR contain 555 and 1164 valid samples, respectively (left column of Table 5). By contrast, because the independent variable FR involves missing values, the two regression equations for simultaneous estimation both contain 555 valid samples (right column). The results of Table 5 show that HD in older adults significantly affect their FR and MR.To compare whether frailty regression and mortality regression should be estimated individually or simultaneously, we use the likelihood ratio test to examine whether the two regression equations are independent from each other. The χ^2^(1) of the likelihood ratio test is 13.983, rejecting that the two regression equations are independent. This implies that recursive simultaneous equations, instead of individual estimation, should be adopted. Additionally, in the individual estimation, the regression coefficient of FR on DR is 0.252, which is considerably smaller than that in simultaneous estimations (1.967). In other words, individual estimation may underestimate the effect of FR on DR.The regression analysis on FR in the panel A of Table 5 shows that TOT_11, DM, Hyper, CTD, and PUD significantly increase FR. By contrast, Alb, TOT_5, and being male significantly reduce FR.The regression analysis on MR in panel B of Table 5 shows that HD significantly increases MR through the recursive effect of FR. Moreover, creatinine (Cr), myocardial infarction (MI), and malignant tumors (M_tumor) directly and significantly increase MR.

## 4. Conclusions and Suggestions

This study employs a database of the Center for Geriatrics and Gerontology at Taichung Veterans General Hospital in Taiwan, which contains the data of 1248 older adults. Furthermore, it uses a simultaneous recursive probit regression model to investigate the association between FR, mortality, and HD in older adults. The research results are as follows: (1) In the likelihood ratio test, when the correlation coefficients of the residual terms of the two regression equations are 0, the χ^2^(1) is 13.983; therefore, the null hypothesis that the two regression equations are independent from each other is significantly rejected. This implies that regression parameters should be determined through recursive simultaneous estimation instead of individual estimation. (2) The regression coefficient for the effect of FR on MR (0.252) obtained in the individual estimation is significantly smaller than that obtained in simultaneous estimation (1.967). This means that individual estimation may underestimate the effect of senior FR on MR. (3) Regarding the existence of FR in older adults, the regression model reveals that TOT_11, DM, Hyper, CTD, and PUD significantly aggravate senior FR, whereas Alb, TOT_5, and being male significantly reduce the level of FR. (4) The level of HD in older adults significantly increases MR through the recursive effect of FR, and Cr, MI, and M_tumor directly and significantly increase MR.

This study finds that HD factors can accelerate older adults’ FR and increase their MR. Therefore, improving older adults’ vitality and reducing the waste of medical resources and air quality are critical topics for Taiwan’s government to alleviate problems related to its aging population. This study suggests that future studies further investigate the difference in HD, FR, and MR between older adults who are hospitalized, living at home, and/or living in care facilities.

## Figures and Tables

**Figure 1 ijerph-17-00211-f001:**
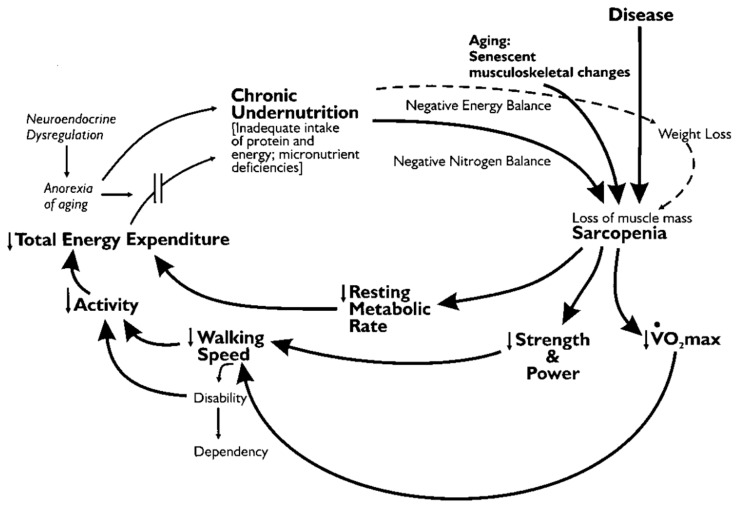
Fried et al.’s cycle of frailty hypothesis regarding frailty (FR). Source: [1] Journal of Gerontology: MEDICAL SCIENCES 2001; 56(3), M147. By permission of Oxford University Press

**Figure 2 ijerph-17-00211-f002:**
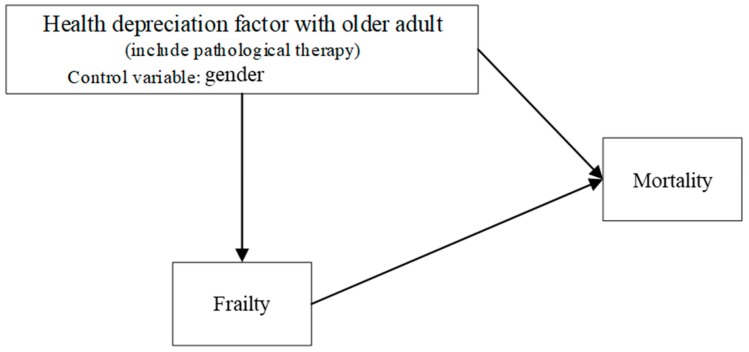
The relationship between health depreciation (HD), FR, and mortality (MR) for older adults.

**Table 1 ijerph-17-00211-t001:** Five dimensions of the FR index.

	Male	Female
Weight Loss	Larger than 5 kg lost unintentionally in prior year	
15 Foot Walk Time	Taking 7 s or more to walk a distance of 6 m	
Grip Strength	grip strength ≤ 24 Kg	grip strength ≤ 18 Kg
Physical Activity (MLTA)	<383 kcal/week	<270 kcal/week
	Kcals per week expended are calculated as follows: 1. walking (MET 2.5 kcal/kg.hr) 2.5′ × weight × time (hours) per day × 7 = a 2. fast walking, climbing stairs (MET 4.5 kcal/kg.hr) 4.5 × weight × time (hours) per day × 7 = b 3. jogging, swimming (MET 6.5 kcal/kg.hr) 6.5 × weight × time (hours)per day × 7 = c 4. almost none (MET 0.5 kcal/kg.hr) 0.5 × weight × time (hours)per day × 7 = d Total energy expenditure per week is adding up a, b, c and d.	
Low Energy/Exhaustion	The question asked is “How often in the last week did you feel this way?” Using the CES–D * Depression Scale, the following two statements: (a) I felt that everything I did was an effort; (b) I could not get going. 0 = rarely or none of the time (<1 day); 1 = some or a little of the time (1–2 days); 2 = a moderate amount of the time (3–4 days); 3 = most of the time. (>4 days) Subjects answering “2” or “3” to either of these questions are categorized as frail by the exhaustion criterion.	

* CES-D = Center for Epidemiologic Studies Depression Scale.

**Table 2 ijerph-17-00211-t002:** Correlation analysis of FR and MR for older adults.

	Relative Health (%)	Frailty (%)	Total
Survival	830 (70.88)	341 (29.12)	1171 (100.00)
Mortality	45 (58.44)	32 (41.56)	77 (100.00)
Total	875 (70.11)	373 (29.89)	1248 (100.00)

Note: Pearson χ^2^(1) = 5.334; *p* = 0.021.

**Table 3 ijerph-17-00211-t003:** Difference analysis of FR and HD.

Variable	Relative Health		Frailty		Number of Samples	Difference Test
	Mean	SD	Mean	SD		
**Panel A: HD (Interval Variables)**						
Albumin (Alb)	3.741	±0.701	3.554	±0.713	925	−3.763 ***
Fasting glucose (F_Glu)	108.782	±29.613	113.905	±32.997	849	2.129 **^a^
Creatinine (Cr)	1.334	±1.168	1.418	±1.280	1164	1.069 ^a^
Glomerular filtration rate (eGFR)	70.495	±34.542	68.540	±34.504	1160	−0.892
Hemoglobin (HgB)	12.328	±2.044	11.863	±1.973	1135	−3.592 ***
Mini nutritional assessment (TOT_5)	1.695	±0.551	1.399	±0.679	1242	−6.930 ***^a^
Modified cumulative illness rating scale―geriatric version (TOT_8)	6.981	±4.001	7.255	±3.737	1228	1.119
Age-adjusted Charlson comorbidity index (TOT_11)	1.237	±1.176	1.649	±1.554	984	4.255 ***^a^
**Panel B: HD (Dummy Variables)**						
Diabetes (DM)	0.362	±0.481	0.485	±0.500	1248	4.059 ***
Hypertension (HT)	0.721	±0.449	0.812	±0.391	1248	3.397 ***
Hyperlipidemia (Hyper)	0.267	±0.443	0.359	±0.480	1248	3.255 ***
Myocardial infarction (MI)	0.088	±0.283	0.105	±0.306	1248	0.922
Congestive heart failure (CHF)	0.168	±0.374	0.212	±0.409	1248	1.838 *
Peripheral artery disease (PAD)	0.075	±0.264	0.091	±0.288	1248	0.936
Cerebrovascular disease (CVAD)	0.179	±0.384	0.231	±0.422	1248	2.087 **
Dementia	0.265	±0.442	0.316	±0.466	1248	1.843 *
Chronic obstructive pulmonary disease (COPD)	0.353	±0.478	0.383	±0.487	1248	1.017
Connective tissue disease (CTD)	0.072	±0.259	0.113	±0.317	1248	2.364 **
Peptic ulcer disease (PUD)	0.232	±0.422	0.284	±0.452	1248	1.954 *
Chronic kidney disease (CKD)	0.386	±0.487	0.504	±0.507	1248	3.854 ***
Hemiparesis (Hemi)	0.023	±0.150	0.040	±0.197	1248	1.699 *
Malignant tumor (M_tumor)	0.039	±0.193	0.027	±0.162	1248	1.056
Moderate-to-severe liver disease (L_disease)	0.138	±0.345	0.177	±0.382	1248	1.751 *
**Panel C: Control Variable**						
Sex (Sex)	0.864	±0.343	0.697	±0.460	1248	6.937 ***

Notes: 1) For interval variables, ^a^ denotes the use of the Satterthwaite–Welch test with unequal variances; 2) for dummy variables, the Mann–Whitney test was conducted; 3) for the control variable, sex was a dummy variable; 4) ***, **, and * denote α = 1%, 5%, and 10%, respectively.

**Table 4 ijerph-17-00211-t004:** Difference analysis of MR and HD.

Variable	Survival		Mortality (MR)		Number of Samples	Difference Test
	Mean	SD	Mean	SD		
**Panel A: HD (Interval Variables)**						
Albumin(Alb)	3.727	±0.672	3.119	±0.903	925	−7.073 ***
Fasting glucose (F_Glu)	110.601	±31.313	106.241	±24.403	849	−1.286 ^a^
Creatinine (Cr)	1.332	±1.635	1.749	±1.635	1164	2.195 **^a^
Glomerular filtration rate (eGFR)	70.186	±33.520	65.724	±46.531	1160	−0.826 ^a^
Hemoglobin (HgB)	12.384	±1.978	10.778	±2.274	1135	−6.346 ***
Mini nutritional assessment (TOT_5)	1.612	±0.591	1.303	±0.749	1242	−81.205 ***^a^
Modified cumulative illness rating scale―geriatric version (TOT_8)	7.082	±3.912	6.770	±4.110	1228	−0.663
Age-adjusted Charlson comorbidity index (TOT_11)	1.342	±1.297	1.826	±1.653	984	2.377 **^a^
**Panel B: HD (Dummy Variables)**						
Diabetes (DM)	0.396	±0.489	0.442	±0.450	1248	0.786
Hypertension (HT)	0.741	±0.438	0.857	±0.352	1248	2.269 **
Hyperlipidemia (Hyper)	0.289	±0.453	0.390	±0.491	1248	1.881 *
Myocardial infarction (MI)	0.086	±0.280	0.208	±0.408	1248	3.581 ***
Congestive heart failure (CHF)	0.175	±0.380	0.273	±0.448	1248	2.155 **
Peripheral artery disease (PAD)	0.079	±0.269	0.104	±0.307	1248	0.792
Cerebrovascular disease (CVAD)	0.190	±0.392	0.273	±0.448	1248	1.784 *
Dementia	0.281	±0.450	0.273	±0.448	1248	0.155
Chronic obstructive pulmonary disease (COPD)	0.358	±0.480	0.429	±0.498	1248	1.251
Connective tissue disease (CTD)	0.085	±0.280	0.065	±0.248	1248	0.626
Peptic ulcer disease (PUD)	0.239	±0.427	0.377	±0.488	1248	2.707 ***
Chronic kidney disease (CKD)	0.413	±0.493	0.545	±0.501	1248	2.273 **
Hemiparesis (Hemi)	0.026	±0.150	0.065	±0.248	1248	2.023 **
Leukemia	0.008	±0.087	0.013	±0.114	1248	0.504
Malignant tumor (M_tumor)	0.030	±0.170	0.117	±0.323	1248	4.007 ***
**Panel C: Control Variable**						
Sex (Sex)	0.816	±0.388	0.792	±0.408	1248	0.487

Notes: 1) For nominal (dummy) variables, the Mann–Whitney test was performed; 2) for interval variables, a denotes the use of the Satterthwaite–Welch test with unequal variances; 3) for the control variable, sex was a dummy variable; 4) ***, **, and * denote α = 1%, 5%, and 10%, respectively.

**Table 5 ijerph-17-00211-t005:** Recursive probit regression analysis for the effects of HD on FR and DR in older adults.

Panel A: FR Regression Model			
Variable	Expected Direction	Individual Estimation	Simultaneous Estimation
Albumin(Alb)	−	−0.029 (−0.30)	−0.114 ** (−1.84)
Fasting glucose (F_Glu)	−	−0.000 (−0.11)	−0.002 (−0.99)
Hemoglobin (HgB)	−	−0.014 (−0.43)	−0.014 (−0.61)
Mini nutritional assessment (TOT_5)	−	−0.202 ** (−2.12)	−0.157 ** (−2.20)
Age-adjusted Charlson comorbidity index (TOT_11)	+	0.060 * (1.40)	0.075 *** (2.71)
Diabetes (DM)	+	0.296 ** (2.29)	0.188 ** (2.07)
Hypertension (HT)	+	0.002 (0.01)	0.091 (0.86)
Hyperlipidemia (Hyper)	+	0.232 ** (1.92)	0.236 *** (3.29)
Congestive heart failure (CHF)	+	0.090 (0.65)	−0.020 (−0.19)
Cerebrovascular disease (CVAD)	−	−0.011 (−0.08)	0.068 (0.80)
Dementia	+	0.204 * (1.71)	0.023 (0.26)
Connective tissue disease (CTD)	+	0.355 ** (2.06)	0.213 * (1.59)
Peptic ulcer disease (PUD)	+	0.036 (0.29)	0.121 * (1.61)
Chronic kidney disease (CKD)	+	0.036 (0.28)	−0.014 (−0.18)
Hemiparesis (Hemi)	+	0.255 (0.90)	−0.090 (−0.31)
Moderate-to-severe liver disease (L_disease)	+	0.098 (0.70)	0.065 (0.72)
Sex	−	−0.442 *** (−3.16)	−0.372 *** (−3.80)
Constant		0.023 (0.05)	0.448 (1.25)
Number of samples		555	555
Likelihood ratio test		χ^2^(17) = 54.10 ***	NA
**Panel B: MR Regression Model**			
Frailty FR	+	0.252 ** (2.09) 0.079 ** (2.05) 0.462 *** (2.85) 0.753 *** (3.36)	1.967 *** (17.59)
Creatinine (Cr)	+		0.048 ** (1.93)
Myocardial infarction (MI)	+		0.421 *** (3.71)
Malignant tumor (M_tumor)	+		0.569 *** (3.19)
Constant	?	−1.820 *** (−18.76)	−1.867 *** (−17.11)
Number of samples		1164	555
Likelihood ratio test		χ^2^(4) = 27.76 ***	χ^2^(21) = 465.83 ***

Likelihood ratio test with a rho of 0: χ^2^(1) = 13.983 ***; Note: Figures in the parentheses are the value of Z; ***, **, and * denote α = 1%, 5%, and 10% (one-tailed), respectively.

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
