# Peer review of "A Study of Frailty, Mortality, and Health Depreciation Factors in Older Adults"

_ijerph, 2019, doi:10.3390/ijerph17010211_

Round 1

Reviewer 1 Report

The study by Lin and colleagues examine the relationship between frailty, mortality, and health depreciation in a sample of older adults from Taiwan. Particularly, this study uses an innovative methodological approach (recursive probit binary regression) to bridge a gap in the literature regarding factors influencing health depreciation and mortality in older adults. The empirical methods are sound and the results indicate a significant association between several health depreciation variables and frailty, as well as between the health depreciation variables and mortality. Specific comments and suggestions are provided below.

Specific Comments:

Abstract:

Page 1, Lines 11-12: “This study finds that a significant positive correlation between...”

Change to “This study finds that a significant positive correlation exists between…” or “This study finds a significant positive correlation between…”

Page 1, Lines 18-19: “…malignant tumors could directly and significantly increased older adults’ mortality.”

Change to “…malignant tumors directly and significantly increased older adults’ mortality.”

Introduction:

Pages 1-2, Lines 44-45: “…fails to consider cognitive emotion and psychological, social, and…”

Tense confusing—are the words “cognitive” and “emotion” separate factors? If so, it should be “…fails to consider cognitive, emotional, psychological, social, and…”

Page 2, Line 47: “…modified it according to the physical characteristics of Asian people.”

Briefly describe some or all of these modifications or refer readers to “see below” (i.e., see section 3.1).

Page 2, Line 70: “…hyperlipidemia, and hyperglycemia,…”

                Remove “and”

Page 3, Lines 89-90: “…a more robust estimation of the empirical analysis.”

Can be shortened to “…a more robust empirical analysis.”

Alternatively, you can directly mention the statistic(s) being estimated (e.g., “…a more robust estimation of the effects of HD on FR and MR in older adults”).

Page 3, Line 91: “…as follows.”

                Change to “…as follows: “

Theoretical Foundation and Empirical Model:

Page 3, Line 99: “…amount of care service provided…”

                Specify how care is quantified (i.e, is this a monetary variable or is it a measure of time?)

Page 3, Line 113: “…in hospital or…”

                Change to “…in hospitals or…”

Page 4, Line 128: “…the number of samples who died (N = 77) accounted for only 6.17% of the total samples (N = 1248)…”

Change to “…the number of individuals who died (N = 77) accounted for only 6.17% of the total sample (N = 1248)…”

Empirical Results Analysis:

Page 4, Line 138: “…as shown in Table 1”

                Missing a period—change to “…as shown in Table 1.”

Pages 4-5, Lines 138-141: “…physical characteristics of Asian people.”

The modifications made to the FR index do not appear to be specific to Asian people (i.e., they may apply to elderly people more generally and are not intrinsically related to being of Asian descent). Instead of referring to these modifications as specific to “Asian people”, the authors might consider presenting them as specific to “the study population” (i.e., elderly people in Taiwan).

Page 5, Line 147: “…that of older adults who survived was 1171.”

                Rephrase—perhaps “…the number of surviving older adults was 1171.”

Page 5, Line 148: “…the correlation between FR and DR is analyzed.”

I cannot find where the abbreviation “DR” is defined. Is it “death rate”? In which case, is it comparable with MR (mortality)?

Conclusions and Suggestions:

Page 8, Line 211: “…establishes a simultaneous recursive probit…”

                Change “establishes” to “utilizes” or “uses”

Author Response

We would like to thank the reviewers for their insightful comments and suggestions. We provide answers point-by-point in the attachments.

Reviewer 2 Report

The subject of the manuscript is of great interest and the study was well designed to attend the objective of the authors. However, small issues should be considered to facilitate the reading of the manuscript.

-The abstract should be written in the past (eg., uses-used; finds-found).

-The authors should standardize the term OLDER ADULTS in the manuscript and not use the term elderly.

- The references are missing in the two first paragraphs of the introduction section.

- The terms regarding the frailty index should be reviewed as pointed out in the text.

- Considering the interval varibles the authors should describe which condition is associted with frailty or mortality (eg., higher age-adjusted Charlston comorbidity index is associated with frailty and not only age-adjusted Charlston comirbidity index; lower albumin levels are associated with frailty and not only Albumin). This should be reviewed in the whole manuscript.

- Also, when describing the interval data items (Page 5, line 155), it should be included the scores of the Mini nutrition assessment, the modified cumulative illness rating scale-geriatric version, and the age-adjusted charlson comorbidity index  and the meaning of them (eg., the score of the Mini nutrition assessment varies from.....to......with lower scores indicating worse nutritional status).

- Tables should be self-explanatory. The use of acronym without a legend makes Table 5 difficult to understand.

Author Response

We would like to thank the reviewers for their comments. Our responses are in the word documents attached.